# The Protective Action of Metformin against Pro-Inflammatory Cytokine-Induced Human Islet Cell Damage and the Mechanisms Involved

**DOI:** 10.3390/cells11152465

**Published:** 2022-08-08

**Authors:** Laura Giusti, Marta Tesi, Federica Ciregia, Lorella Marselli, Lorenzo Zallocco, Mara Suleiman, Carmela De Luca, Silvia Del Guerra, Mariachiara Zuccarini, Marco Trerotola, Decio L. Eizirik, Miriam Cnop, Maria R. Mazzoni, Piero Marchetti, Antonio Lucacchini, Maurizio Ronci

**Affiliations:** 1School of Pharmacy, University of Camerino, 62032 Camerino, Italy; 2Department of Clinical and Experimental Medicine, University of Pisa, 56126 Pisa, Italy; 3Laboratory of Rheumatology, GIGA Research, CHU de Liège, University of Liège, 4000 Liège, Belgium; 4Department of Pharmacy, University of Pisa, 56126 Pisa, Italy; 5Center for Advanced Studies and Technologies (CAST), University of Chieti-Pescara, 66100 Chieti, Italy; 6Department of Medical, Oral and Biotechnological Sciences, University “G. d’Annunzio” of Chieti-Pescara, 66100 Chieti, Italy; 7ULB Center for Diabetes Research, Université Libre de Bruxelles, 1070 Brussels, Belgium; 8Department of Pharmacy, University “G. d’Annunzio” of Chieti-Pescara, 66100 Chieti, Italy

**Keywords:** β-cell, cytokines, metformin, proteomics, label-free shotgun analysis

## Abstract

Metformin, a drug widely used in type 2 diabetes (T2D), has been shown to protect human β-cells exposed to gluco- and/or lipotoxic conditions and those in islets from T2D donors. We assessed whether metformin could relieve the human β-cell stress induced by pro-inflammatory cytokines (which mediate β-cells damage in type 1 diabetes, T1D) and investigated the underlying mechanisms using shotgun proteomics. Human islets were exposed to 50 U/mL interleukin-1β plus 1000 U/mL interferon-γ for 48 h, with or without 2.4 µg/mL metformin. Glucose-stimulated insulin secretion (GSIS) and caspase 3/7 activity were studied, and a shotgun label free proteomics analysis was performed. Metformin prevented the reduction of GSIS and the activation of caspase 3/7 induced by cytokines. Proteomics analysis identified more than 3000 proteins in human islets. Cytokines alone altered the expression of 244 proteins (145 up- and 99 down-regulated), while, in the presence of metformin, cytokine-exposure modified the expression of 231 proteins (128 up- and 103 downregulated). Among the proteins inversely regulated in the two conditions, we found proteins involved in vesicle motility, defense against oxidative stress (including peroxiredoxins), metabolism, protein synthesis, glycolysis and its regulation, and cytoskeletal proteins. Metformin inhibited pathways linked to inflammation, immune reactions, mammalian target of rapamycin (mTOR) signaling, and cell senescence. Some of the changes were confirmed by Western blot. Therefore, metformin prevented part of the deleterious actions of pro-inflammatory cytokines in human β-cells, which was accompanied by islet proteome modifications. This suggests that metformin, besides use in T2D, might be considered for β-cell protection in other types of diabetes, possibly including early T1D.

## 1. Introduction

Diabetes mellitus (DM) is a disorder of the metabolism of carbohydrate, fat, and protein, due to the interplay of genetic and environmental factors [1,2]. It is characterized by an absolute or relative shortage of insulin production and secretion by the pancreatic islet β-cells [1,3,4]. In 2021 there were 537 million people (age 20–79 yrs) with DM, which is expected to increase to 643 million by 2030 and 783 million by 2045 [2]. Morbidity and mortality in diabetic subjects are high, mainly due to the acute metabolic and chronic vascular complications of the disease, and it has been calculated that approximately 6.7 million diabetic adults died in 2021 [2]. In parallel, the direct costs of diabetes grew to USD 966 billion in 2021 [2]. Therefore, better strategies to prevent and treat this disease are needed.

Type 2 diabetes (T2D) is the most common form of DM, representing approximately 90% of all cases [1,2,3,4]. Several drugs are used to treat T2D, and metformin is the most widely employed [5,6]. Metformin is derived from galegine, a natural component of *Galega Officinalis*, a plant used in herbal medicine in medieval Europe and that was introduced into clinical use for the treatment of T2D in the 1950s [7]. Although its molecular mechanisms of action remain to be fully elucidated, metformin has been proven to be a safe and effective therapy, and it is now recommended as the first-line pharmacological treatment against T2D [8]. Metformin reduces blood glucose levels by decreasing hepatic glucose production, modifying the gut microbiome, and enhancing GLP-1 secretion [9,10,11,12]. In addition, the drug has anti-inflammatory properties, as indicated by its reduction of the neutrophil to lymphocyte ratio in subjects with T2D, anti-oxidative stress action, direct inhibitory effects on NF-kB signaling, and suppression of inflammatory cytokines in non-diabetic individuals [13,14,15].

Cytokines are small proteins produced by immune cells and other cell types that may have pro-inflammatory or anti-inflammatory effects, and which act via autocrine, paracrine, and/or endocrine mechanisms. A large body of evidence shows that pro-inflammatory cytokines (locally produced by immune cells in the course of insulitis) are involved in the pathogenesis of type 1 diabetes (T1D) [16,17,18,19,20,21]. At the level of the β-cells, they contribute to β-cell dysfunction and/or death during the early (particularly type 1 interferons, such as interferon-a (IFN-a)) and late (particularly interleukin-1b (IL-1b) and interferon-g (IFN-g)) phases of insulitis in type 1 diabetes.

Interestingly, a few studies have shown that metformin could have direct protective effects on β-cells under metabolic stress, including non-diabetic and T2D human islet cells [22,23,24,25,26,27,28,29,30]. However, it is not known whether metformin directly protects human β-cells against the damage induced by pro-inflammatory cytokines, nor the mechanisms possibly involved. Previous studies have indicated that the drug could shelter chondrocytes from IL-1β injury [31], reduce cytokine production in the cardiac muscle following ischemia-reperfusion [32], and limit the damage induced by lipopolysaccharide exposure in human bronchial epithelial cells [33].

In the present study we evaluated if metformin can defend human islet cells from IL-1β + IFN-γ-induced dysfunction and death. The mechanisms possibly involved were investigated at the proteome level with the use of shotgun proteomics, a bottom-up technique that enables comprehensive protein identification and profiling [34]. Although different tissues and cell types have been extensively evaluated using this approach [35,36], shotgun proteomics have been used to analyze pancreatic islets, the key tissue in diabetes pathogenesis [37,38,39], in only a few studies [40,41,42].

We show that, in our experimental conditions, metformin was able to shield isolated human islets from part of the insults induced by the tested cytokines, which was associated with several changes at the proteomic level, with the involvement of pathways mainly regulating inflammation and oxidative stress. 

## 2. Methods

### 2.1. Human Pancreatic Islets

Isolated islets were prepared by enzymatic digestion and gradient purification from the pancreas of 14 multiorgan donors (age: 71 ± 9 years; 5M/9F; BMI: 26 ± 3 kg/m^2^) [43], with written consent by next-of-kin. Glands that were not suitable for clinical purposes were processed [43,44] with the approval of the local Ethics Committee (#2615 of 15 January 2014). We selected donors without a known history of diabetes. Following isolation, islets were cultured in M199 medium (Euroclone SpA, Milan, Italy) containing 5.5 mM glucose, supplemented with 10% (*v*/*v*) adult bovine serum, 100 U/mL penicillin, 100 μg/mL streptomycin, 50 μg/mL gentamicin, and 750 ng/mL amphotericin B (all from Sigma-Aldrich, St. Louis, MO, USA) at 37 °C in a CO_2_ incubator. For the purpose of the present study, approximately 1000 islets were incubated with cytokines (50 U/mL IL-1β, 1000 U/mL IFN-γ) for 48 h, in the presence and absence of metformin (2.4 µg/mL) (Sigma-Aldrich). This is a therapeutic concentration of the drug, which has been used in our laboratory previously [28,29]. The cytokine concentrations were based on those used by us and others in previous experiments [17,45,46,47]. Afterwards, isolated islets were used for functional, survival, and proteomics analyses, as described below.

### 2.2. Insulin Secretion Studies

Insulin release experiments were conducted as previously described [43,48,49]. After 45 min pre-incubation at 3.3 mM glucose, batches of 15 handpicked islets were challenged acutely (45 min) with 3.3- and 16.7-mM glucose. Then the islets were subjected to acid-alcohol extraction for insulin content measurement, as previously reported [43,48,49]. Insulin was quantified using a radioimmunometric assay (DIAsource ImmunoAssays S.A., Nivelles, Belgium). Insulin release was expressed as a percentage of the total insulin content. Insulin stimulation index was calculated as the ratio of insulin release at 16.7 mM glucose over the release at 3.3 mM glucose.

### 2.3. Caspase 3/7 Activity Assay

A Caspase-Glo^®^ 3/7 assay kit (Promega Corporation, Madison, WI, USA) was used to detect caspase 3/7 activity, as described in [50,51]. Briefly, batches of 10 size-matched islets were seeded in a white solid 96-well plate, in a total volume of 100 μL/well. Then 100 μL of caspase 3/7 reagent, a solution containing luciferase and a tetrapeptide substrate linked to aminoluciferin, was added to each well and incubated for 1 h at room temperature. Following caspase cleavage of substrate, aminoluciferin was released and processed by luciferase, resulting in the production of light. Luminescence was recorded with a FLUOstar Omega microplate reader (BMG Labtech, Ortenberg, Germany).

### 2.4. Protein Extraction from Human Pancreatic Islets

The proteomic analysis was performed with islet preparations obtained from three different multiorgan donors (representing the biological replicates). Protein extraction from human pancreatic islets was performed as previously described [52]. Briefly, isolated islets were collected and washed twice with PBS (37 °C). Cells were suspended in the rehydration solution (7 M urea, 2 M thiourea, 4% CHAPS, 60 mM dithiothreitol (DTT), 0.002% bromophenol blue) containing 50 mM NaF, 2 mM Na_3_VO_4_, 1 μL/10^6^ cells of protease inhibitors, 1 µM trichostatin A, and 10 mM nicotinamide. After stirring and sonication (4 s, 5 times) cells were allowed to rehydrate for 1 h at room temperature (RT) with occasional stirring. Thereafter, the solution was centrifuged at 17,000× *g* for 5 min at RT. The protein concentration of the resulting supernatant was determined using the Bio-Rad RC/DC-protein assay (Bio-Rad). BSA was used as a standard.

### 2.5. Protein Fractionation

For shotgun analysis [53], technical triplicate experiments were performed on each of the three human islet preparations. For each preparation, three different conditions were analyzed, i.e., control islets, cytokine alone treatment, and cytokine plus metformin treatment. For this purpose, approximately 1000 human islets were treated as described above, and protein extracts were processed as follows: aliquots (40 µg of proteins) were loaded onto 12% acrylamide resolving gel and subjected to 1D-electrophoresis, as previously performed by us and others [43,54]. After protein staining using Coomassie blue R-250, 16 gel bands, matched for each lane, were excised and washed twice with wash buffer (25 mM NH_4_HCO_3_ in 50% acetonitrile). Afterwards, proteins were reduced with 10 mM dithiothreitol (45 min, 56 °C) and alkylated with 55 mM iodoacetamide (30 min at RT in the dark). After two washes with the washing buffer, protein bands were completely dried in a centrivap vacuum centrifuge. Then the dried pieces of gel were rehydrated for 30 min at 4 °C in a porcine trypsin (Promega, Madison, WI, USA) solution (3 ng/μL in 100 mM NH_4_HCO_3_) and incubated overnight at 37 °C. The reaction was quenched by adding 10% trifluoroacetic acid. The samples were stored at −20 °C before being analyzed by LC-MS/MS.

### 2.6. Shotgun Label Free Analysis

The resulting peptides, 48 samples for each subject (16 controls, 16 treated with cytokines, and 16 treated with cytokines + metformin), were grouped by band and analyzed in technical triplicates using LC-MS/MS using a Proxeon EASY-nLCII (Thermo Fisher Scientific, Milan, Italy) chromatographic system coupled to a Maxis HD UHR-TOF (Bruker Daltonics GmbH, Bremen, Germany) mass spectrometer equipped with a nanoESI spray source. Peptides were loaded on an EASY-Column C18 trapping column (2 cm L, 100 µm I.D, 5 µm ps, Thermo Fisher Scientific) and subsequently separated on an Acclaim PepMap100 C18 (75 µm I.D., 25 cm L, 5 µm ps, Thermo Fisher Scientific) nano scale chromatographic column. The flow rate was set to 300 nL/min, and the gradient (mobile phase A: 0.1% formic acid in H_2_O) was from 3 to 35% of mobile phase B (1% formic acid in acetonitrile) in 80 min, followed by 35 to 45% in 10 min and from 45 to 90% in 11 min. The mass spectrometer was operated in positive ion polarity and Auto MS/MS mode (data dependent acquisition—DDA), using N2 as a collision gas for CID fragmentation. Precursors in the range of 350 to 2200 m/z (excluding 1220.0–1224.5 m/z) with a preferred charge state +2 to +5 (excluding singly charged ions) and absolute intensity above 4706 counts were selected for fragmentation in a maximum cycle time of 3 s. After acquiring one MS/MS spectrum, the precursors were actively excluded from selection for 30 s. Isolation width and collision energy for MS/MS fragmentation were set according to the mass and charge state of the precursor ions (from 3 to 9 Da and from 21 eV to 55 eV). In-source reference lock mass (1221.9906 *m*/*z*) was acquired online throughout the runs. Altogether, 432 instrumental runs were performed. Each raw data file was converted to mzXML format and submitted to LFQ processing (see below).

### 2.7. Raw Data Processing and Quantitative Analysis

Raw mass spectrometry data were analyzed using the PEAKS^®^ Studio 7.5 software using the “correct precursor only” option. Spectra were matched against the neXtProt database (including isoforms as of June 2017; 42,151 entries), and the false discovery rate (FDR) was set to 0.1% at the peptide-spectrum matches (PSM) level. The post-translational modification (PTM) profile was set as follows: fixed cysteine carbamidomethylation (ΔMass: 57.02), variable methionine oxidation (ΔMass: 15.99), and glutamine and asparagine deamidation (ΔMass: 0.98). Non-specific cleavage was allowed to one end of the peptides, with a maximum of 2 missed cleavages and trypsin enzyme specificity. The highest error mass tolerances for precursors and fragments were set at 10 ppm and 0.05 Da, respectively. After processing every single raw data point, the label free quantification (LFQ) tool of PEAKS Studio was used to detect differentially expressed proteins. Parameters for LFQ were set as follows: quantification type as label free quantification; mass error tolerance, 10.0 ppm; retention time shift tolerance, 2.0 min; FDR threshold, 0.5%. The nine samples for each of the sixteen slices were allotted to 3 groups corresponding to Ctrl, Cyt, and Cyt + Met. For quantitative analysis, the significance threshold at the protein level was set to ≥ 20−10lgP with a fold change ≥2.0. Sixteen lists of differentially expressed proteins were obtained for each subject.

### 2.8. Pathway Analysis

Gene ontology and pathway analyses of differentially expressed proteins were performed with Metascape v3.5 (https://metascape.org/, accessed on 7 July 2022) [55] and Ingenuity Pathway Analysis (IPA, QIAGEN Redwood City, www.qiagen.com/ingenuity, Build version: 321,501 M, Content version: 21249400, accessed on 7 July 2022), respectively. IPA core analysis provides not only gene functional annotation, canonical pathway, and network discovery, but also estimates the status of upstream regulators and downstream effects associated with canonical pathways, diseases, and functions. The upstream regulator analysis highlights the expected effects between the transcriptional regulators and their target genes [56]. The predicted activation or inhibition of each transcriptional regulator is inferred by the z score, which in turn is derived from the protein ratios in the dataset (z scores >2.0 indicate that a molecule is activated, whereas z scores <−2.0 indicate the inhibition of target molecules). SwissProt accession numbers with corresponding ratios were imported into the software, and the analysis was performed selecting only direct relationships among genes and molecules in all species and confidence settings were set to high predicted or experimentally observed. An IPA comparison analysis between the results of the different sections for every condition was also performed.

### 2.9. Western Blot

Western blot (WB) was performed as described [52], in order to confirm the shotgun results for ERAP2 and IFI30. Aliquots of protein samples (20 μg and 50 μg for IFI30 and ERAP2, respectively) were mixed with Laemmly solution, run in 4–15% polyacrylamide gels (Mini-PROTEAN^®^ Precast Gels, Biorad) using a mini-Protean Tetracell (Biorad), and transferred onto nitrocellulose membranes (0.2 μm) using a Trans-Blot Turbo transfer system (Biorad), as described [52]. Anti-ERAP2 mouse monoclonal antibody (R&D Systems, Inc, Minneapolis, MN, USA) was used at 1:500 dilution, whereas anti-IFI30 mouse monoclonal antibody (sc-393507, Santa-Cruz Biotechnology, Inc, Dallas, TX, USA) was diluted 1:500. β-actin was used as a reference, and the anti-β-actin mouse monoclonal antibody (1:1000 dilution) (Merck group, De) was applied. HRP-goat anti-mouse secondary antibody was used at 1:10,000 dilution. Immunoblots were developed using the enhanced chemiluminescence detection system (ECL). Chemiluminescent images were acquired using LAS4010 (GE Health Care). The optical density (OD) of specific immunoreactive bands was quantified using Image Quant-L software (GE Health Care) and normalized by β-actin.

### 2.10. Statistical Analysis

Data are expressed as means ± SEM. The ANOVA test followed by Tukey correction was applied to assess the difference between groups in the insulin secretion and caspase activation experiments. A *p* value less than 0.05 was considered statistically significant. For proteomics analyses, we used the results generated with the islet preparations obtained from three different multiorgan donors (biological replicates), with peptides from each fraction analyzed in technical triplicates. The false discovery rate (FDR) was set to 0.1% at the peptide–spectrum matches (PSM) level, resulting in an average protein FDR lower than 1.0%. Expression analysis for the relative abundance of identified proteins was performed at the band-slice level using the label-free quantification module PEAKS-Q, part of PEAKS Studio v. 7.5. This quantification method is based on the MS1 ion peak intensity of the extracted chromatograms of peptides detected in multiple samples and applies an expectation-maximization algorithm to detect and resolve overlapping features. A high-performance retention time alignment algorithm was also used to align features of the same peptide from multiple samples. The significance of the LFQ proteomics data provided directly from the software PEAKS Studio was calculated using the PEAKS Q method, which is similar to the significance B, as previously defined [57]. Briefly, protein ratios are calculated as the median of peptide ratios, minimizing the effect of outliers and normalizing the protein ratios, to correct for unequal protein amounts. An outlier significance score for log protein ratios is computed and a *p*-value for detection of significant outlier ratios is defined. Peptide ratios are calculated using the XIC of three different peptides. The differentially expressed proteins were calculated for each band using 0.5% PSMs FDR, and the resulting ID lists were filtered, considering only peptides confidently identified in at least 1 sample with significance ≥20 (−10lgP) and quality factor ≥0.5, and by considering only proteins identified with a significance ≥20 and fold change ≥2. The lists of the resulting dysregulated proteins of each band–slice were merged and manually curated to remove proteins identified in multiple adjacent bands, non-distinguishable isoforms, and keratins.

## 3. Results

### 3.1. Effects of Metformin on Cytokine-Induced Damage

The first aim of the present work was to assess whether metformin could protect human islets from cytokine toxicity, as previously observed for lipotoxic and glucotoxic damage [28,30]. The insulin content in cytokine-exposed islets was lower than in control cells and was not significantly modified by the presence of metformin (Figure 1A). As expected, the insulin stimulation index in response to glucose was significantly reduced after cytokine treatment (Figure 1B). However, with metformin added to the cytokines, the β-cell responsiveness to glucose stimulation was comparable to that of control islets (Figure 1B). As shown in Figure 1C, the presence of metformin led to a significant decline of cytokine-induced caspase 3/7 activity, a marker of cell apoptosis. These results show that metformin could counteract part of the deleterious actions of proinflammatory cytokines on human islet cells.

### 3.2. Identification of Islet Proteins Using Multidimensional Shotgun Proteomics 

Isotope free shotgun proteomics analysis, performed after 1D-PAGE separation, was carried out in control and treated islets, to assess and identify differentially expressed proteins in cytokine- and cytokine plus metformin-exposed vs. control samples. A global view of the experimental workflow is shown in Figure 2A. In 1D gels, the 16 extracted bands were highlighted and paired to the corresponding average mass of the proteins identified in each band (Figure 2B). By merge-processing the 16 bands of each gel lane, altogether 3115 proteins were identified (Appendix A): 1857 in subject 1; 2471 in subject 2; and 2585 in subject 3. The proteins identified across all samples were 1525 and those present in at least two preparations were 2271. Figure 2C shows a Venn diagram of the three series.

We then assessed the proteins significantly affected by the addition of cytokines and cytokines plus metformin. Taking into consideration the proteins present in at least two preparations, we found 244 proteins significantly affected by cytokines (145 up- and 99 down-regulated), of which 32 were exclusively detected in the cytokine-treated islets (Appendix A). The addition of metformin to cytokine treatment significantly altered the expression of 231 proteins (128 up- and 103 downregulated), compared to control samples (Appendix A). Of these, 19 were only detected in the cytokine-treated islets.

As shown in Figure 3A, 212 differentially expressed proteins were found in common between the two different comparisons (Table 1), mostly regulated in the same direction (98 up–up and 88 down–down, Figure 3B and Appendix A). Interestingly, 26 proteins showed an inverse regulation compared to control islets (Appendix A). Of these, 11 were downregulated by cytokine treatment and upregulated after the addition of metformin (Figure 3B). Most proteins were involved in vesicle motility (transgelin, Ras-related protein Rab-14), defense against oxidative stress (peroxiredoxins, PRDX2 and PRDX5) and metabolism (flavin reductase, mitochondrial ATP synthase subunit O). Among the 15 proteins upregulated by cytokines and downregulated by metformin (Figure 3B), we detected proteins involved in protein synthesis (40S ribosomal proteins S3, S6, S9, eukaryotic translation initiation factor 4E), glycolysis, or glycolysis regulation (triosephosphate isomerase, pyruvate kinase); protein folding and secretion (peptidyl-prolyl cis-trans isomerase FKBP2, protein disulfide-isomerase); and cytoskeletal proteins or proteins interacting with the cytoskeleton (myosin light polypeptide 6, Ras-related protein Ral-A, coactosin-like protein).

### 3.3. Gene Ontology, Analysis of Canonical Pathway, and Upstream Regulators

To investigate the biological meaning of the above-mentioned proteomics changes, the curated lists of differentially expressed proteins, namely cytokines vs. control and cytokines plus metformin vs. control conditions were submitted to gene ontology enrichment analysis (https://metascape.org/, accessed on 7 July 2022) [55]. The enrichment analysis results for the biological process and reactome curated pathways are shown in Figure 4. The most significantly enriched pathway includes 57 proteins annotated as “Cytokine Signaling in Immune system”, with a logP score Enrichment of −39.89. GO results for the biological processes annotation indicates that most proteins were related to the immune system response and response to cytokines. The above-described overlap of the proteins modulated by cytokines and cytokines plus metformin (n: 212) led to the similarity of the GO-enriched terms in the two conditions. Therefore, to better discriminate the effects of metformin in the presence of pro-inflammatory cytokines in human islets, ingenuity pathway analysis (IPA) was used. Table 2 shows the top 25 canonical pathways obtained from protein differentially expressed in cytokine-treated samples compared to the control. As expected, most of them resulted as associated with inflammatory processes. Those with the highest *p*-values were linked to the “Antigen Presentation Pathway”, “Phagosome Maturation”, “Acute Phase Response Signaling”, and “Interferon Signaling”. Other inflammation-related pathways such as the “IL-17 Signaling” and “Neuroinflammation Signaling”; and some stress-related pathways, such as the “NRF2-mediated Oxidative Stress Response”, “Mitochondrial Disfunction”; and “Type 1 Diabetes Mellitus Signaling” also resulted as significantly enriched. The upstream regulators resulting from IPA confirmed the activation of several transcription factors, such as STAT1, IRF1, RELA, and IRF7, with high positive values of z-score. In addition, a significant inhibition of MAPK1 was shown. The top 25 upstream regulators are listed in Table 3.

We then assessed if and how the addition of metformin to cytokines modified the above-described scenario. Figure 5A shows the heatmap of the topmost significantly affected canonical pathways, when comparing the two different treatments. A positive or negative z-score value indicates the activation or inhibition of canonical pathways, significant for a value >2 and <−2, respectively. Metformin reduced cytokine-promoted activation of IL-6, IL-8, and IL-15 signaling; JAK/Stat signaling; mTOR signaling; senescence pathway; HMGB1 signaling; and Gα_12/13_ signaling. Glycolysis I was also reduced in the presence of metformin. Concerning the upstream regulators, metformin addition reduced the activation induced by cytokines of thrombospondin (THBS4), interleukin 15 (IL-15), interferon epsilon (IFNE), and tumor necrosis factor ligand superfamily member 12 (TNFSF12) and inhibited regulatory-associated protein of mTOR (RAPTOR) (Figure 5B).

### 3.4. Western Blot (WB) Analysis of ERAP2 and IFI30 in Human Islets

The different expression of IFI30 and ERAP2 (proteins involved in antigen processing) in human islets treated with cytokines and cytokines plus metformin was also evaluated using WB analysis. A specific 28 KDa immunoreactive band was detected for IFI30 (Figure 6A), while four specific immunoreactive bands, with apparent weights of 110, 105, 70, and 45 kDa (the first 2 reported in Figure 6A), were detected for ERAP2, corresponding to different isoforms of this protein. In our shotgun experiments, ERAP2 was identified in band 3 of the 1DE gel, corresponding to the highest molecular weight isoforms 1 and 3 of 110 and 105 KDa, respectively. According to the shotgun proteomic analysis, WB showed a significantly higher expression of both IFI30 and ERAP2 in islets treated with cytokines alone than in control samples (Figure 6B,C). The changes induced by the addition of metformin, as observed by proteomic evaluation (significant decrease of IFI30 and significant increase of ERAP2 expression), remained as an apparent, although not significant, trend.

## 4. Discussion

The present study reports the analysis of human pancreatic islets after 48 h exposure to pro-inflammatory cytokines (50 U/mL IL-1β and 1000 U/mL IFN-γ), with or without the presence of metformin, using a multidimensional shotgun proteomics approach. We observed protective effects of metformin on citokine-induced β-cell functional damage and increased capase 3/7 activity and investigated the underlying molecular mechanisms by assessing the related proteome modulation.

The previously observed deleterious effects of cytokines on human islet β-cell function and survival [21,58] were confirmed in the present study. Interestingly, the presence of metformin partially prevented β-cell dysfunction and activation of caspase 3/7 (a marker of apoptosis), similarly to the protective action that the compound exerts on human islets exposed to lipoglucotoxicity [28,30] and on islets isolated from type 2 diabetic patients [29]. This suggests that metformin has broad beneficial effects on stressed human islet cells, regardless of the insulting condition, possibly due to pleiotropic, and so far poorly understood, mechanisms.

A few previous studies have investigated the effects of pro-inflammatory cytokines on human islet gene and protein expression [45,46,59,60,61]. Recently, Nakayasu et al. used tandem mass tags and the label-free technique to study islet proteomics in depth [42]. In our study, 3115 proteins were identified in control samples (untreated islets), of which 3014 were also reported by Nakayasu et al. [42], indicating the good reproducibility of the results between the two approaches. After cytokine exposure, we found that 244 proteins were differentially expressed compared with the control islets, mainly pertaining to the cytoskeleton, immune response signaling, apoptosis signaling, energy metabolism, protein metabolism, and RNA metabolism. Of them, 57 were also reported in ref [42]. All but two of these 57 proteins were modulated in the same direction in both studies and were mostly upregulated compared to control islets.

Interestingly, the addition of metformin to cytokine-treated islets inhibited several canonical pathways and upstream regulators related to inflammation, such as interleukin and HMGB1 (high mobility group protein B1) signaling and IL15 [62,63]. An anti-inflammatory effect of metformin has been described in a few models, including pancreatic islets [64,65,66,67]. The mechanisms by which metformin dampens inflammation are still unclear. However, the drug can reduce oxidative stress [29,68], which is linked to the promotion of inflammatory processes [69,70]. Accordingly, in metformin-treated islets, we also found a significantly increased expression of proteins involved in the defense against oxidative stress [71,72], such as glutathione S-transferase α1 and 2 (GSTA1 and 2); thioredoxin 1 (TRX1); and peroxiredoxins 2, 3, and 5 (PRDX2, 3, and 5). The peroxiredoxin/thioredoxin antioxidant system has been described in rodent β-cell lines and pancreatic islets as a relevant protective mechanism against oxidative damage [73,74,75,76]. The specific role of each peroxiredoxin is still debated. PRXD1 has been recently described as the main isoform involved in protection against hydrogen peroxide and peroxynitrite [77]. Interestingly, we presently observed that the two mitochondrial peroxiredoxin isoforms, PRDX3 and PRDX5, resulted as upregulated in metformin-treated islets, suggesting that the mechanism of metformin action in mitochondria could go beyond the inhibition of complex I [78].

Metformin was also able to reduce cytokine-induced immune response through the inhibition of pathways such as the systemic lupus erythematosus in T cell signaling pathway, PKCθ signaling in T lymphocytes, role of NFAT in regulation of the immune response, iCOS-iCOSL signaling in T helper cells, and acute phase response signaling. Some immune-suppressive action of metformin via modulation of T lymphocyte activity has been described [79]. In our study, the addition of metformin to cytokines reduced the expression of components of major histocompatibility complex (MHC) class I antigen presentation, such as HLA class I histocompatibility antigen B alpha chain and C alpha chain (HLA-B and HLA-C) and beta-2-microglobulin (B2M), which is critical for the expression of functional HLA-I molecules on the cell surface [80,81]. These changes might have protective effects on β-cells in T1D, by decreasing the presentation of b-cell β-cell neoantigens for the immune system [82]. Conversely, the expression of HLA-A and G α chains increased in the islets exposed to cytokines and metformin. HLA-G, a non-classical HLA class I molecule having immunomodulatory properties, has previously been found to be constitutively expressed in human pancreatic islets [83] and has been reported to be involved in the attenuation of autoimmune and inflammatory processes [84].

Our proteomic data also showed variations in the expression of IFI30 and ERAP2, enzymes involved in antigen processing [85,86] and that are required for peptide binding to MHC class I antigens and for generating MHC class II- restricted epitopes from disulfide bond-containing proteins, respectively. IFI30 has been implicated in the pathogenesis of some autoimmune diseases, and genetic polymorphisms of ERAP2 and its paralog ERAP1 are associated with increased susceptibility to autoimmune/chronic inflammatory disorders [87,88]. In agreement with findings previously reported with different models [89,90], we observed that cytokines induced overexpression of both enzymes in human islets, which was confirmed by Western blot analysis. Interestingly, the shotgun proteomics results showed significantly decreased expression of IFI30 and increased expression of ERAP2 when metformin was added to the cytokines, which was tendentially confirmed by Western blot analysis.

Among the proteins exclusively upregulated in metformin-treated islets, calcium/calmodulin-dependent serine protein kinase (CASK) and transcription factor JUNB deserve a special mention, since previous work has implicated them in protecting β cells against cytokines-induced damage [91,92]. CASK, which plays a role in synaptic transmembrane protein anchoring and ion channel trafficking, seems to be also involved in insulin secretion from β cells [93]. It has been reported that IL-1β reduces CASK expression in INS-1 cells and rat islets, while its overexpression counteracts the cytokine-induced β cell dysfunction, by improving insulin secretion [91]. Although, in our experiments, we did not detect a decrease of CASK expression following cytokine exposure, the addition of metformin induced the expression of this protein kinase, suggesting a possible protective mechanism of the drug at this level. Furthermore, it has been shown that the proinflammatory cytokines IL-1β and INF-γ induce an initial and transient upregulation of JUNB in INS-1E cells [92] and that JUNB overexpression reduces cytokine-induced β cell death [94]. Our data therefore suggest that metformin could contribute to keeping active a self-defense system that is otherwise transient.

Another effect of metformin on cytokine-treated islets inferred from our proteomics results is the inhibition of the upstream regulator RAPTOR (regulatory-associated protein of mTOR) and hence the mTOR (serine/threonine-protein kinase mTOR) signaling pathway. It is known that metformin downregulates mTOR signaling through either 5′-adenosine monophosphate–activated protein kinase (AMPK)-dependent- or independent mechanisms [95,96]. We found a decreased expression of proteins involved in mTOR activity, specifically eukaryotic translation initiation factor 4E (eIF-4E), a regulator of translation, and the 40S ribosomal proteins S3, S6, and S9, involved in protein synthesis [97]. Interestingly, mTOR inhibition is associated with increased autophagic fluxes [98], which, in turn, could favor the function and survival of stressed β-cells [99,100,101,102].

Of interest, the senescence pathway was also activated by cytokine-treatment and inhibited by metformin addition. The beneficial action of metformin in mitigating aging hallmarks has previously been reported [95]. Notably, cellular senescence has been identified as a key process in both T1D and T2D development [103,104,105]. In a murine model of pf T2D, it was found that insulin resistance accelerates β-cell senescence, while removal of senescent β-cells (senolysis) improves β-cell function and glucose homeostasis, leading to a better disease outcome [103]. Moreover, it has been reported that during T1D development a subset of β-cells apparently acquire the senescent phenotype and thus contribute to immune-mediated β-cell destruction [104].

This study has some limitations. Although the concentration of metformin that we used is in the therapeutic range, the actual levels of the drug in the pancreatic islet microenvironment in vivo are currently unclear. In addition, pancreatic islets are heterogeneous within the same pancreas and between subjects [38,106], and it is unknown if cytokines and metformin have different effects on islets from different individuals. Nevertheless, the islet functional and proteomics results were generated with cutting edge methodologies applied in experienced laboratories and analyzed using strict statistical assessment. This allowed confirming and supporting data from previous studies and, more importantly, to add new knowledge to the field.

In conclusion, the present study shows, for the first time, that metformin prevents, at least in part, the deleterious actions of pro-inflammatory cytokines on human β-cell function and islet cell caspase 3/7 activation, which is accompanied by several modifications of the islet proteome. These modifications include pathways involved in inflammation, immunity, mTOR signaling, and cellular senescence, all known to impact β-cell health. This evidence suggests that metformin, a widely used drug for the treatment of T2D, might be repurposed for β-cell protection in early T1D.

## Figures and Tables

**Figure 1 cells-11-02465-f001:**
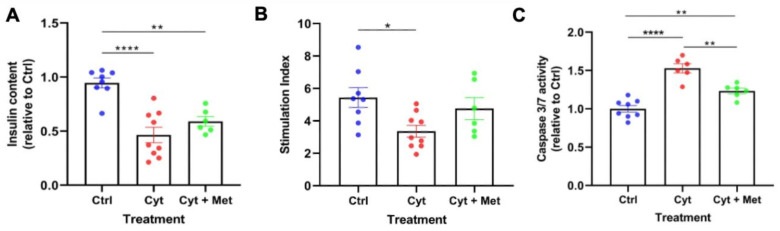
Effect of metformin on cytokine-induced β-cell damage. (**A**) Insulin content, reduced after cytokine-treatment, was marginally affected by metformin. (**B**) Insulin stimulation index was reduced after 48 h treatment with cytokines (Cyt) and tended to return to the control values (Ctrl) in islets treated with metformin (Cyt + Met). (**C**) Cytokines induced a significant activation of caspase 3/7, while metformin significantly reduced this activation. One to three replicates from three to four independent islet preparations were studied. The different groups were compared with One-way ANOVA followed by the Tukey correction. **** *p* < 0.0001, ** *p* < 0.01, * *p* < 0.05.

**Figure 2 cells-11-02465-f002:**
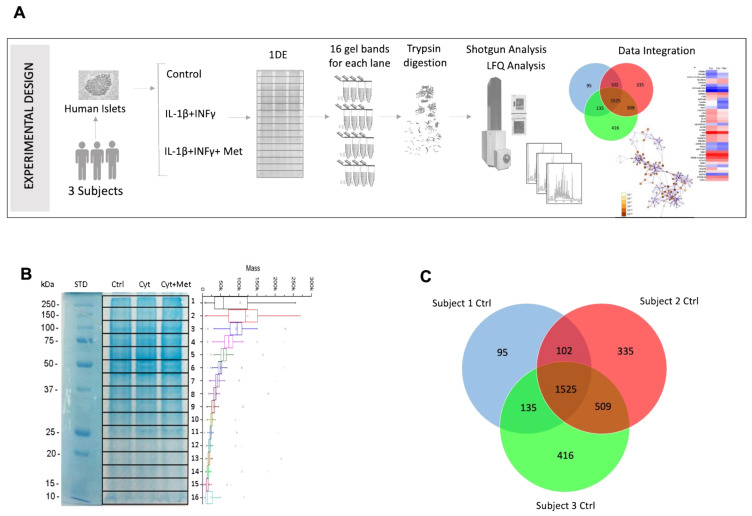
Global view of the proteomics analysis of human pancreatic islets treated with cytokines in the absence and in the presence of metformin using shotgun proteomics. (**A**) Experimental design of the proteomics analysis of pancreatic islets. (**B**) A representative 1DE gel in which the 16 extracted bands are highlighted and paired to the corresponding average mass of the proteins identified in each band. (**C**) Venn diagram showing the number of proteins found in the control islets.

**Figure 3 cells-11-02465-f003:**
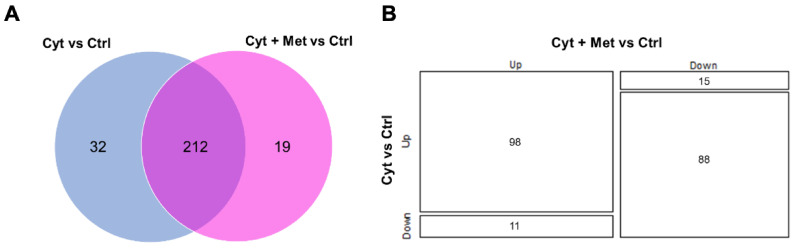
Comparison of differentially expressed proteins in cytokines vs. control and cytokines plus metformin vs. control. (**A**) Venn diagram showing the number of differentially expressed proteins in common between Cyt vs. Ctrl and Cyt + Met vs. Ctrl. (**B**) Mosaic plot showing the directionality of the significantly affected proteins in common between the two different comparisons.

**Figure 4 cells-11-02465-f004:**
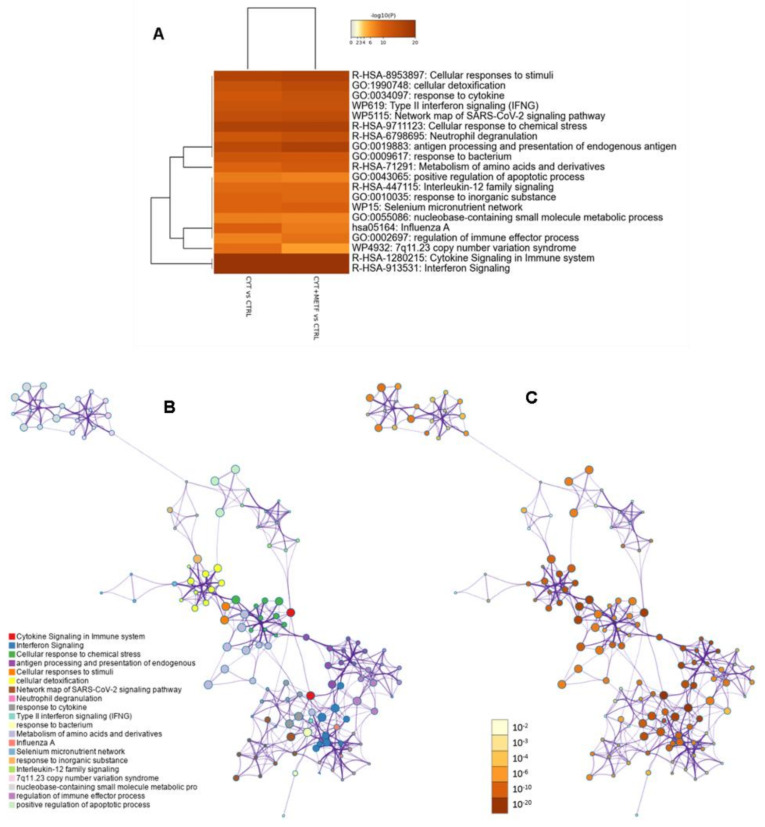
Enrichment analysis results for biological process. (**A**) All statistically enriched terms were identified and used for filtering, after computing the accumulative hypergeometric *p*-values and enrichment factors. The remaining significant terms were then hierarchically clustered into a tree based on Kappa-statistical similarities among their gene memberships. Then a 0.3 kappa score was applied as a threshold to divide the tree into term clusters. The term with the best *p*-value within each cluster was selected as its representative term and displayed in a dendrogram. The heatmap cells are colored by their *p*-values. (**B**) A subset of representative terms from the full cluster were selected and converted into a network layout. Each term is represented by a circle node, where its size is proportional to the number of input genes fall into that term, and its color represent its cluster identity (i.e., nodes of the same color belong to the same cluster). Terms with a similarity score >0.3 are linked by an edge (the thickness of the edge represents the similarity score). The network was visualized with Cytoscape (v3.1.2) with “force-directed” layout and with edges bundled for clarity. One term from each cluster was selected to have its term description shown as a label. (**C**) The same enrichment network has its nodes colored by *p*-value, as shown in the legend. The darker the color, the more statistically significant the node is.

**Figure 5 cells-11-02465-f005:**
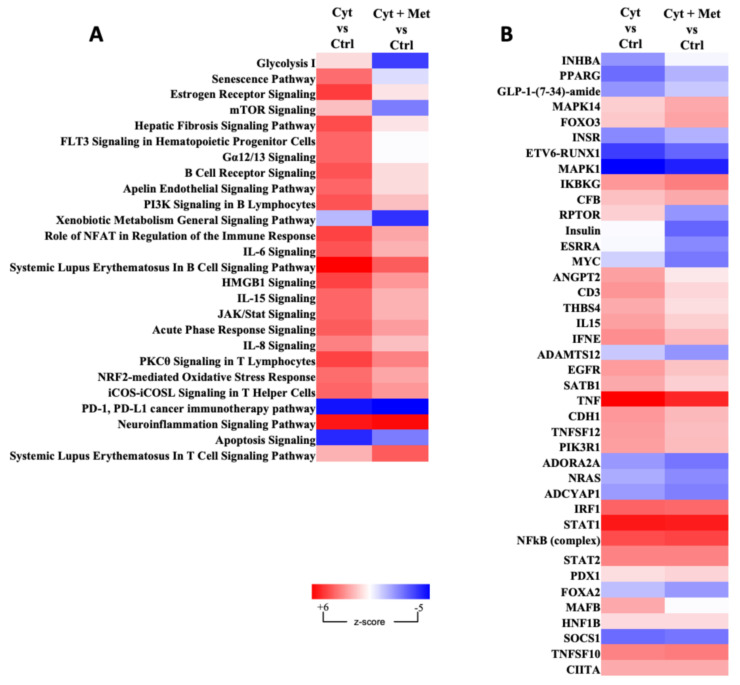
Heat map showing the canonical pathways and upstream regulators differentially regulated by cytokines and cytokines plus metformin vs. the respective control samples. Activated (red) or inhibited (blue) canonical pathways (**A**) and upstream regulators (**B**) obtained by IPA analysis of human islet differentially regulated proteins in the two different comparisons. The brighter the color, the more intense the change is. Canonical pathways (panel **A**) were selected based on the highest differences in z-score values.

**Figure 6 cells-11-02465-f006:**
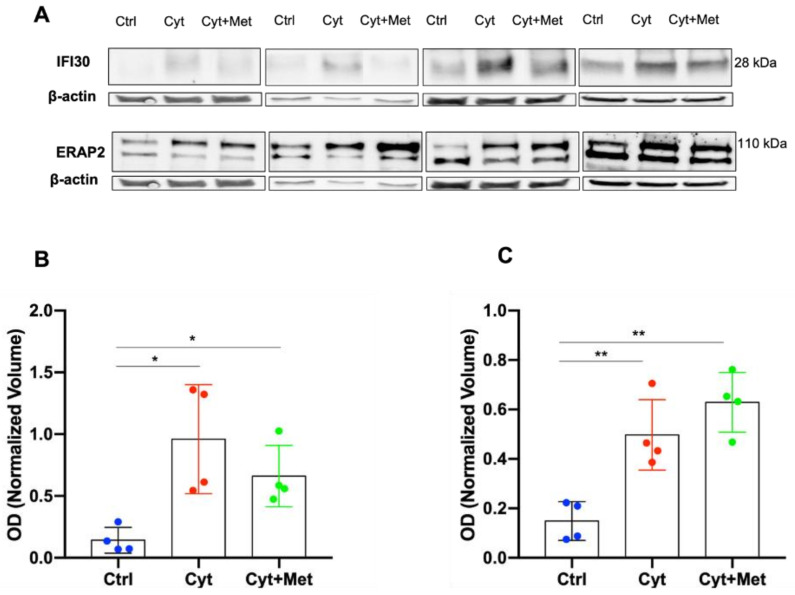
Western blot (WB) analysis was used to evaluate the expression of IFI30 and ERAP2, in both cytokines and cytokines + metformin treated islets compared to control samples. (**A**) Immunoreactive bands of IFI30 (28KDa), ERAP2 (110KDa), and β-actin, as observed in four independent islet preparations in the three different conditions. (**B**,**C**) Violin plots of the normalized OD, reported as mean values ± SEM, for IFI30 (panel **B**) and ERAP2 (panel **C**). The dashed line represents the median values, and the dot lines represent the first and the third quartiles. Statistical analysis was performed using a parametric paired *t* test. * *p* < 0.05, ** *p* < 0.01.

**Table 1 cells-11-02465-t001:** List of proteins significantly modulated by both cytokines and cytokines plus metformin vs. the respective control samples.

ID	Protein Name	Gene	Cyt/Ctrl	Cyt + Met/Ctrl
	**Apoptosis**			
O95140	MITOFUSIN-2 (PTHR10465:SF1)	*MFN2*	0.01	0.01
Q9H1Y0	AUTOPHAGY PROTEIN 5 (PTHR13040:SF2)	*ATG5*	0.01	0.01
Q9NR28	DIABLO HOMOLOG, MITOCHONDRIAL (PTHR32247:SF3)	*DIABLO*	0.23	0.95
Q13813	SPECTRIN ALPHA CHAIN, NON-ERYTHROCYTIC 1 (PTHR11915:SF427)	*SPTAN1*	0.54	0.15
Q13501	SEQUESTOSOME-1 (PTHR15090:SF0)	*SQSTM1*	2.06	1.8
	**Cytoskeleton, vesicle motility/intracellular transport/vesicle release**			
P13637	SODIUM/POTASSIUM-TRANSPORTING ATPASE SUBUNIT ALPHA-3 (PTHR43294:SF15)	*ATP1A3*	0.01	0.01
Q5D862	FILAGGRIN-2 (PTHR22571:SF24)	*FLG2*	0.01	0.01
Q6KB66	KERATIN, TYPE II CYTOSKELETAL 80 (PTHR45616:SF1)	*KRT80*	0.01	0.01
Q86YZ3	HORNERIN (PTHR22571:SF25)	*HRNR*	0.01	0.27
Q8IV36	PROTEIN HID1 (PTHR21575:SF12)	*HID1*	0.01	0.01
Q8N1N4	KERATIN, TYPE II CYTOSKELETAL 78 (PTHR45616:SF18)	*KRT78*	0.01	0.01
Q99442	TRANSLOCATION PROTEIN SEC62 (PTHR12443:SF9)	*SEC62*	0.01	0.01
Q01995	TRANSGELIN (PTHR18959:SF40)	*TAGLN*	0.09	1.11
P61204	ADP-RIBOSYLATION FACTOR 3 (PTHR11711:SF316)	*ARF3*	0.1	0.98
Q06141	REGENERATING ISLET-DERIVED PROTEIN 3-ALPHA (PTHR22803:SF123)	*REG3A*	0.18	0.12
O94875	SORBIN AND SH3 DOMAIN-CONTAINING PROTEIN 2 (PTHR14167:SF56)	*SORBS2*	0.19	0.2
Q9BVK6	TRANSMEMBRANE EMP24 DOMAIN-CONTAINING PROTEIN 9 (PTHR22811:SF37)	*TMED9*	0.2	0.85
P08670	VIMENTIN (PTHR45652:SF5)	*VIM*	0.32	0.32
P61106	RAS-RELATED PROTEIN RAB-14 (PTHR24073:SF185)	*RAB14*	0.35	1.59
Q14019	COACTOSIN-LIKE PROTEIN (PTHR10829:SF29)	*COTL1*	1.14	0.4
Q969Q5	RAS-RELATED PROTEIN RAB-24 (PTHR24073:SF471)	*RAB24*	1.89	2.53
P07737	PROFILIN-1 (PTHR13936:SF14)	*PFN1*	4.98	17.49
Q9BQE5	APOLIPOPROTEIN L2 (PTHR14096:SF27)	*APOL2*	6.47	5.93
	*actin and actin related protein or actin-binding protein*			
Q9BYX7	BETA-ACTIN-LIKE PROTEIN 3-RELATED (PTHR11937:SF278)	*POTEKP*	0.01	0.01
P68032	ACTIN, ALPHA CARDIAC MUSCLE 1 (PTHR11937:SF176)	*ACTC1*	0.32	0.09
P60660	MYOSIN LIGHT POLYPEPTIDE 6 (PTHR23048:SF7)	*MYL6*	1.69	0.43
P63261	ACTIN, CYTOPLASMIC 2 (PTHR11937:SF414)	*ACTG1*	2	2.15
O43795	UNCONVENTIONAL MYOSIN-IB (PTHR13140:SF277)	*MYO1B*	2.98	3
	*intercellular signal molecule*			
P02751	FIBRONECTIN (PTHR19143:SF267)	*FN1*	0.01	0.01
O15240	NEUROSECRETORY PROTEIN VGF (PTHR15159:SF2)	*VGF*	1.76	1.94
	*tubulin*			
Q9NRH3	TUBULIN GAMMA-2 CHAIN (PTHR11588:SF79)	*TUBG2*	0.01	0.01
Q71U36	TUBULIN ALPHA-1A CHAIN (PTHR11588:SF133)	*TUBA1A*	1.27	
	*small GTPase*			
P62826	GTP-BINDING NUCLEAR PROTEIN RAN (PTHR24071:SF23)	*RAN*	0.39	1.48
P11233	RAS-RELATED PROTEIN RAL-A (PTHR24070:SF174)	*RALA*	1.98	0.62
P11234	RAS-RELATED PROTEIN RAL-B (PTHR24070:SF199)	*RALB*	5.02	2.86
	**Defense repair/immune response/Signaling in Immune system**			
P05451	LITHOSTATHINE-1-ALPHA (PTHR22803:SF105)	*REG1A*	0.42	0.27
P10599	THIOREDOXIN (PTHR10438:SF18)	*TXN*	1.79	2.26
Q06323	PROTEASOME ACTIVATOR COMPLEX SUBUNIT 1 (PTHR10660:SF5)	*PSME1*	1.99	1.78
Q9Y6N5	SULFIDE:QUINONE OXIDOREDUCTASE, MITOCHONDRIAL (PTHR10632:SF2)	*SQOR*	2.1	3.46
O15533	TAPASIN (PTHR23411:SF22)	*TAPBP*	3.97	6.54
P13164	INTERFERON-INDUCED TRANSMEMBRANE PROTEIN 1 (PTHR13999:SF6)	*IFITM1*	4.8	4.98
P05362	INTERCELLULAR ADHESION MOLECULE 1 (PTHR13771:SF9)	*ICAM1*	24.83	23.94
P00751	COMPLEMENT FACTOR B (PTHR46393:SF1)	*CFB*	32.89	29.22
P14174	MACROPHAGE MIGRATION INHIBITORY FACTOR (PTHR11954:SF6)	*MIF*	34.9	51.02
P05161	UBIQUITIN-LIKE PROTEIN ISG15 (PTHR10666:SF267)	*ISG15*	45.7	52.83
Q10589	BONE MARROW STROMAL ANTIGEN 2 (PTHR15190:SF1)	*BST2*	56.28	55.56
O14879	INTERFERON-INDUCED PROTEIN WITH TETRATRICOPEPTIDE REPEATS 3 (PTHR10271:SF3)	*IFIT3*	69.33	64.83
P09913	INTERFERON-INDUCED PROTEIN WITH TETRATRICOPEPTIDE REPEATS 2 (PTHR10271:SF4)	*IFIT2*	100	100
Q96AZ6	INTERFERON-STIMULATED GENE 20 KDA PROTEIN (PTHR12801:SF59)	*ISG20*	100	96.33
	*peroxidase*			
P30048	THIOREDOXIN-DEPENDENT PEROXIDE REDUCTASE, MITOCHONDRIAL (PTHR42801:SF4)	*PRDX3*	0.18	0.95
P32119	PEROXIREDOXIN-2 (PTHR10681:SF161)	*PRDX2*	0.34	1.02
P30044	PEROXIREDOXIN-5, MITOCHONDRIAL (PTHR10430:SF16)	*PRDX5*	0.46	1.8
P30041	PEROXIREDOXIN-6 (PTHR43503:SF11)	*PRDX6*	0.79	0.14
Q06830	PEROXIREDOXIN-1 (PTHR10681:SF111)	*PRDX1*	2.2	1.86
	*oxidase/oxidoreductase*			
O14618	COPPER CHAPERONE FOR SUPEROXIDE DISMUTASE (PTHR10003:SF88)	*CCS*	0.23	0.01
Q9NRD8	DUAL OXIDASE 2 (PTHR11972:SF67)	*DUOX2*	5.2	3.26
P04179	SUPEROXIDE DISMUTASE [MN], MITOCHONDRIAL (PTHR11404:SF6)	*SOD2*	6.39	5.12
P00450	CERULOPLASMIN (PTHR11709:SF226)	*CP*	21.87	21.33
P13284	GAMMA-INTERFERON-INDUCIBLE LYSOSOMAL THIOL REDUCTASE (PTHR13234:SF8)	*IFI30*	38.02	6.53
	*heterotrimeric G-protein*			
P32456	GUANYLATE-BINDING PROTEIN 2 (PTHR10751:SF112)	*GBP2*	64.5	69
Q96PP8	GUANYLATE-BINDING PROTEIN 5 (PTHR10751:SF40)	*GBP5*	69.41	55.56
P32455	GUANYLATE-BINDING PROTEIN 1 (PTHR10751:SF96)	*GBP1*	72	72.8
Q96PP9	GUANYLATE-BINDING PROTEIN 4 (PTHR10751:SF17)	*GBP4*	100	100
	*chemokine*			
P09341	GROWTH-REGULATED ALPHA PROTEIN (PTHR10179:SF69)	*CXCL1*	11.21	21.38
P78556	C-C MOTIF CHEMOKINE 20 (PTHR12015:SF108)	*CCL20*	57.22	55.56
P19875	C-X-C MOTIF CHEMOKINE 2 (PTHR10179:SF80)	*CXCL2*	58.95	70
P02778	C-X-C MOTIF CHEMOKINE 10 (PTHR10179:SF47)	*CXCL10*	100	100
	*membrane traffic protein*			
Q03169	TUMOR NECROSIS FACTOR ALPHA-INDUCED PROTEIN 2 (PTHR21292:SF4)	*TNFAIP2*	4.78	6.27
P20591	INTERFERON-INDUCED GTP-BINDING PROTEIN MX1 (PTHR11566:SF51)	*MX1*	39.72	43.43
	*ATP-binding cassette (ABC) transporter*			
Q03518	ANTIGEN PEPTIDE TRANSPORTER 1 (PTHR24221:SF249)	*TAP1*	10.59	9.41
Q03519	ANTIGEN PEPTIDE TRANSPORTER 2 (PTHR24221:SF237)	*TAP2*	13.9	11.72
	**Cytokine and Interferon Signaling in Immune system**			
P10145	INTERLEUKIN-8 (PTHR10179:SF42)	*CXCL8*	17.68	19.64
	*major histocompatibility complex protein*			
P04439	HLA CLASS I HISTOCOMPATIBILITY ANTIGEN, A ALPHA CHAIN (HUMAN LEUKOCYTE ANTIGEN A) (HLA-A)	*HLA-A*	0	100
P61769	BETA-2-MICROGLOBULIN (PTHR19944:SF62)	*B2M*	2.74	1.68
P01911	HLA CLASS II HISTOCOMPATIBILITY ANTIGEN, DRB1-15 BETA CHAIN (PTHR19944:SF99)	*HLA-DRB1*	14.79	15.21
P10321	HLA CLASS I HISTOCOMPATIBILITY ANTIGEN, C ALPHA CHAIN (HLA-C) (HLA-CW)	*HLA-C*	37.18	11.22
P01903	HLA CLASS II HISTOCOMPATIBILITY ANTIGEN, DR ALPHA CHAIN (PTHR19944:SF63)	*HLA-DRA*	51.82	51.11
P01889	HLA CLASS I HISTOCOMPATIBILITY ANTIGEN, B ALPHA CHAIN	*HLA-B*	100	0
	**Metabolism**			
P09601	HEME OXYGENASE 1 (PTHR10720:SF1)	*HMOX1*	0.01	0.01
P34059	N-ACETYLGALACTOSAMINE-6-SULFATASE (PTHR42693:SF33)	*GALNS*	0.01	0.01
P16233	PANCREATIC TRIACYLGLYCEROL LIPASE (PTHR11610:SF115)	*PNLIP*	0.21	0.38
P04054	PHOSPHOLIPASE A2 (PTHR11716:SF94)	*PLA2G1B*	0.33	0.01
Q8NCW5	NAD(P)H-HYDRATE EPIMERASE (PTHR13232:SF11)	*NAXE*	0.41	0.12
Q9H2U2	INORGANIC PYROPHOSPHATASE 2, MITOCHONDRIAL (PTHR10286:SF49)	*PPA2*	0.69	0.11
P47985	CYTOCHROME B-C1 COMPLEX SUBUNIT RIESKE, MITOCHONDRIAL-RELATED (PTHR10134:SF20)	*UQCRFS1*	1.55	2.83
P15954	CYTOCHROME C OXIDASE SUBUNIT 7C, MITOCHONDRIAL (PTHR13313:SF1)	*COX7C*	2.42	3.84
P19971	THYMIDINE PHOSPHORYLASE (PTHR10515:SF0)	*TYMP*	5.68	6.01
	*ATP synthase*			
P06576	ATP SYNTHASE SUBUNIT BETA, MITOCHONDRIAL (PTHR15184:SF58)	*ATP5F1B*	0.28	0.52
P48047	ATP SYNTHASE SUBUNIT O, MITOCHONDRIAL (PTHR11910:SF1)	*ATP5PO*	0.65	1.11
	*kinase*			
P07205	PHOSPHOGLYCERATE KINASE 2 (PTHR11406:SF10)	*PGK2*	0.01	0.01
Q6ZUJ8	PHOSPHOINOSITIDE 3-KINASE ADAPTER PROTEIN 1 (PTHR16267:SF12)	*PIK3AP1*	2.89	2.79
P14618	PYRUVATE KINASE PKM (PTHR11817:SF15)	*PKM*	6.41	0.47
P32189	GLYCEROL KINASE (PTHR10196:SF56)	*GK*	52.08	52.06
	*dehydrogenase*			
P15559	NAD(P)H DEHYDROGENASE [QUINONE] 1 (PTHR10204:SF1)	*NQO1*	0.31	0.25
P04406	GLYCERALDEHYDE-3-PHOSPHATE DEHYDROGENASE (PTHR10836:SF111)	*GAPDH*	0.33	0.86
Q8NBQ5	ESTRADIOL 17-BETA-DEHYDROGENASE 11 (PTHR24322:SF489)	*HSD17B11*	0.5	0.04
P40926	MALATE DEHYDROGENASE, MITOCHONDRIAL (PTHR11540:SF16)	*MDH2*	0.51	0.15
P47989	XANTHINE DEHYDROGENASE/OXIDASE (PTHR11908:SF80)	*XDH*	60.25	62.5
P14902	INDOLEAMINE 2,3-DIOXYGENASE 1 (PTHR28657:SF2)	*IDO1*	100	100
	*isomerase*			
Q13907	ISOPENTENYL-DIPHOSPHATE DELTA-ISOMERASE 1 (PTHR10885:SF5)	*IDI1*	0.5	0.01
P60174	TRIOSEPHOSPHATE ISOMERASE (PTHR21139:SF24)	*TPI1*	1.29	0.18
	*transferase*			
P51580	THIOPURINE S-METHYLTRANSFERASE (PTHR10259:SF11)	*TPMT*	0.01	0.01
P08263	GLUTATHIONE S-TRANSFERASE A1 (PTHR11571:SF157)	*GSTA1*	0.29	0.62
P09210	GLUTATHIONE S-TRANSFERASE A2 (PTHR11571:SF243)	*GSTA2*	0.31	0.62
Q99735	MICROSOMAL GLUTATHIONE S-TRANSFERASE 2 (PTHR10250:SF13)	*MGST2*	0.36	0.21
P24752	ACETYL-COA ACETYLTRANSFERASE, MITOCHONDRIAL (PTHR18919:SF156)	*ACAT1*	0.43	0.86
Q14749	GLYCINE N-METHYLTRANSFERASE (PTHR16458:SF2)	*GNMT*	0.44	0.01
P40261	NICOTINAMIDE N-METHYLTRANSFERASE (PTHR10867:SF37)	*NNMT*	1.99	2.17
P43490	NICOTINAMIDE PHOSPHORIBOSYLTRANSFERASE (PTHR43816:SF1)	*NAMPT*	2.33	2.55
P21980	PROTEIN-GLUTAMINE GAMMA-GLUTAMYLTRANSFERASE 2 (PTHR11590:SF6)	*TGM2*	3.39	2.62
	*reductase*			
P30043	FLAVIN REDUCTASE (NADPH) (PTHR43355:SF2)	*BLVRB*	0.36	1.45
P52895	ALDO-KETO REDUCTASE FAMILY 1 MEMBER C1-RELATED (PTHR11732:SF153)	*AKR1C2*	2.87	2.21
	**Protein synthesis, modification, folding, secretion, degradation**			
Q32P28	PROLYL 3-HYDROXYLASE 1 (PTHR14049:SF5)	*P3H1*	0.01	0.01
Q96JB6	LYSYL OXIDASE HOMOLOG 4 (PTHR45817:SF5)	*LOXL4*	0.01	0.01
P61278	SOMATOSTATIN (PTHR10558:SF2)	*SST*	0.26	0.01
P16870	CARBOXYPEPTIDASE E (PTHR11532:SF62)	*CPE*	0.59	0.42
P10909	CLUSTERIN (PTHR10970:SF1)	*CLU*	0.63	0.21
P07237	PROTEIN DISULFIDE-ISOMERASE (PTHR18929:SF101)	*P4HB*	1.2	0.42
P01275	GLUCAGON (PTHR11418:SF0)	*GCG*	1.9	0.73
P26885	PEPTIDYL-PROLYL CIS-TRANS ISOMERASE FKBP2 (PTHR45779:SF3)	*FKBP2*	1.98	0.67
P28062	PROTEASOME SUBUNIT BETA TYPE-8 (PTHR11599:SF53)	*PSMB8*	2.06	2.47
P11021	ENDOPLASMIC RETICULUM CHAPERONE BIP (PTHR19375:SF380)	*HSPA5*	2.18	1.39
Q00653	NUCLEAR FACTOR NF-KAPPA-B P100 SUBUNIT (DNA-BINDING FACTOR KBF2) (H2TF1)]	*NFKB2*	3.28	2.9
Q9UL46	PROTEASOME ACTIVATOR COMPLEX SUBUNIT 2 (PTHR10660:SF6)	*PSME2*	3.7	3.51
P10645	CHROMOGRANIN-A (CgA)	*CHGA*	3.86	5.24
P29590	PROTEIN PML (PTHR25462:SF241)	*PML*	4.64	4.39
P28065	PROTEASOME SUBUNIT BETA TYPE-9 (PTHR11599:SF50)	*PSMB9*	6.02	5.32
P40306	PROTEASOME SUBUNIT BETA TYPE-10 (PTHR11599:SF41)	*PSMB10*	10.69	13.67
P01833	POLYMERIC IMMUNOGLOBULIN RECEPTOR (PTHR11860:SF82)	*PIGR*	19	15.97
O15205	UBIQUITIN D (PTHR47731:SF1)	*UBD*	62.5	55.63
	*metalloprotease*			
Q99797	MITOCHONDRIAL INTERMEDIATE PEPTIDASE (PTHR11804:SF5)	*MIPEP*	0.01	0.01
Q6P179	ENDOPLASMIC RETICULUM AMINOPEPTIDASE 2 (PTHR11533:SF239)	*ERAP2*	3.56	35.71
P28838	CYTOSOL AMINOPEPTIDASE (PTHR11963:SF39)	*LAP3*	6.61	7.38
	*cysteine protease*			
P09936	UBIQUITIN CARBOXYL-TERMINAL HYDROLASE ISOZYME L1 (PTHR10589:SF19)	*UCHL1*	0.62	0.22
P25774	CATHEPSIN S (PTHR12411:SF525)	*CTSS*	4.99	5.02
	*serine protease*			
Q9ULP0	PROTEIN NDRG4 (PTHR11034:SF21)	*NDRG4*	0.01	0.01
P07478	TRYPSIN-2 (PTHR24264:SF53)	*PRSS2*	0.36	0.31
P07477	TRYPSIN-1 (PTHR24264:SF59)	*PRSS1*	0.44	0.37
Q6GPI1	CHYMOTRYPSINOGEN B2 (PTHR24250:SF53)	*CTRB2*	0.46	0.46
	*ribosomal protein*			
P52815	39S RIBOSOMAL PROTEIN L12, MITOCHONDRIAL (PTHR45987:SF4)	*MRPL12*	0.01	0.01
P05387	60S ACIDIC RIBOSOMAL PROTEIN P2 (PTHR21141:SF5)	*RPLP2*	0.35	0.32
Q96A35	39S RIBOSOMAL PROTEIN L24, MITOCHONDRIAL (PTHR12903:SF0)	*MRPL24*	0.4	0.01
Q9H2W6	39S RIBOSOMAL PROTEIN L46, MITOCHONDRIAL (PTHR13124:SF12)	*MRPL46*	0.48	0.02
Q9H9J2	39S RIBOSOMAL PROTEIN L44, MITOCHONDRIAL (PTHR11207:SF5)	*MRPL44*	0.59	0.14
P62857	40S RIBOSOMAL PROTEIN S28 (PTHR10769:SF3)	*RPS28*	0.61	0.45
P62753	40S RIBOSOMAL PROTEIN S6 (PTHR11502:SF16)	*RPS6*	1.42	0.92
P23396	40S RIBOSOMAL PROTEIN S3 (PTHR11760:SF36)	*RPS3*	1.57	0.32
P60866	40S RIBOSOMAL PROTEIN S20 (PTHR11700:SF8)	*RPS20*	1.87	1.31
P46781	40S RIBOSOMAL PROTEIN S9 (PTHR11831:SF23)	*RPS9*	2.19	0.96
	*ubiquitin-protein ligase*			
Q99816	TUMOR SUSCEPTIBILITY GENE 101 PROTEIN (PTHR23306:SF17)	*TSG101*	0.35	1.38
O14933	UBIQUITIN/ISG15-CONJUGATING ENZYME E2 L6 (PTHR24068:SF43)	*UBE2L6*	11.67	10.77
	**RNA/tRNA synthesis/metabolism/turnover**			
Q13123	PROTEIN RED (PTHR12765:SF7)	*IK*	0.01	0.01
Q9NR30	NUCLEOLAR RNA HELICASE 2 (PTHR47958:SF109)	*DDX21*	0.01	0.01
Q96EP5	DAZ-ASSOCIATED PROTEIN 1 (PTHR48027:SF12)	*DAZAP1*	0.2	0.49
Q9H583	HEAT REPEAT-CONTAINING PROTEIN 1 (PTHR13457:SF1)	*HEATR1*	2.24	7.58
P23381	TRYPTOPHAN-TRNA LIGASE, CYTOPLASMIC (PTHR10055:SF1)	*WARS1*	18.27	21.83
P55769	NHP2-LIKE PROTEIN 1 (PTHR23105:SF38)	*SNU13*	23.22	51.02
Q9UI30	MULTIFUNCTIONAL METHYLTRANSFERASE SUBUNIT TRM112-LIKE PROTEIN (PTHR12773:SF2)	*TRMT112*	51.09	50.85
P62304	SMALL NUCLEAR RIBONUCLEOPROTEIN E (PTHR11193:SF9)	*SNRPE*	51.39	50.61
Q9Y6K5	2’-5’-OLIGOADENYLATE SYNTHASE 3 (PTHR11258:SF4)	*OAS3*	52.63	52.68
	*translation factor*			
Q53EL6	PROGRAMMED CELL DEATH PROTEIN 4 (PTHR12626:SF3)	*PDCD4*	0.32	0.32
P06730	EUKARYOTIC TRANSLATION INITIATION FACTOR 4E (PTHR11960:SF14)	*EIF4E*	1.67	0.68
P68104	ELONGATION FACTOR 1-ALPHA 1-RELATED (PTHR23115:SF222)	*EEF1A1*	2.11	2.37
	*Chromatin remodeling and histone*			
P84243	HISTONE H3.3-RELATED (PTHR11426:SF228)	*H3-3A*	0.79	
P62805	HISTONE H4 (PTHR10484:SF163)	*H4-16*	2	2.45
Q16777	HISTONE H2A TYPE 2-C (PTHR23430:SF130)	*H2AC20*	2.75	1.25
P0C0S5	HISTONE H2A.Z (PTHR23430:SF47)	*H2AZ1*	34.01	51.25
Q7L7L0	HISTONE H2A TYPE 3 (PTHR23430:SF220)	*H2AW*	51.25	50.64
	*DNA binding*			
P43246	DNA MISMATCH REPAIR PROTEIN MSH2 (PTHR11361:SF35)	*MSH2*	0.01	0.01
P12004	PROLIFERATING CELL NUCLEAR ANTIGEN (PTHR11352:SF5)	*PCNA*	0.81	0.25
P42224	SIGNAL TRANSDUCER AND ACTIVATOR OF TRANSCRIPTION 1-ALPHA/BETA (PTHR11801:SF18)	*STAT1*	9.25	9.63
	*exocytosis*			
P13521	SECRETOGRANIN-2 (PTHR15119:SF0)	*SCG2*	0.45	0.21
	**Miscellaneus**			
Q12860	CONTACTIN-1 (PTHR44170:SF10)	*CNTN1*	0.01	0.01
Q14966	ZINC FINGER PROTEIN 638 (PTHR15592:SF1)	*ZNF638*	0.01	0.01
Q5BJF2	SIGMA INTRACELLULAR RECEPTOR 2 (PTHR31204:SF1)	*TMEM97*	0.01	0.01
Q969Z3	MITOCHONDRIAL AMIDOXIME REDUCING COMPONENT 2 (PTHR14237:SF27)	*MTARC2*	0.01	0.01
Q9BZH6	WD REPEAT-CONTAINING PROTEIN 11 (PTHR14593:SF5)	*WDR11*	0.01	0.01
Q8WY91	THAP DOMAIN-CONTAINING PROTEIN 4 (PTHR15854:SF4)	*THAP4*	0.17	50.62
Q9Y2W1	THYROID HORMONE RECEPTOR-ASSOCIATED PROTEIN 3 (PTHR15268:SF16)	*THRAP3*	0.41	0.22
P05408	NEUROENDOCRINE PROTEIN 7B2 (PTHR12738:SF0)	*SCG5*	0.44	0.01
P48059	LIM AND SENESCENT CELL ANTIGEN-LIKE-CONTAINING DOMAIN PROTEIN 1 (PTHR24210:SF11)	*LIMS1*	0.45	0.41
O14967	CALMEGIN (PTHR11073:SF7)	*CLGN*	0.49	0.17
P05060	SECRETOGRANIN-1 (PTHR10583:SF4)	*CHGB*	0.52	0.37
P02766	TRANSTHYRETIN (PTHR10395:SF12)	*TTR*	0.61	0.26
Q9UHG3	PRENYLCYSTEINE OXIDASE 1 (PTHR15944:SF3)	*PCYOX1*	0.85	0.21
O75323	PROTEIN NIPSNAP HOMOLOG 2 (PTHR21017:SF14)	*NIPSNAP2*	0.96	0.22
Q96C19	EF-HAND DOMAIN-CONTAINING PROTEIN D2 (PTHR13025:SF2)	*EFHD2*	1.57	0.77
Q12907	VESICULAR INTEGRAL-MEMBRANE PROTEIN VIP36 (PTHR12223:SF36)	*LMAN2*	2.14	2.75
Q6UVJ0	SPINDLE ASSEMBLY ABNORMAL PROTEIN 6 HOMOLOG (PTHR44281:SF1)	*SASS6*	2.35	3.32
Q9UKY7	PROTEIN CDV3 HOMOLOG (PTHR16284:SF13)	*CDV3*	2.57	1.63
Q9C002	NORMAL MUCOSA OF ESOPHAGUS-SPECIFIC GENE 1 PROTEIN (PTHR14256:SF3)	*NMES1*	2.61	3.12
Q8N339	METALLOTHIONEIN-1M (PTHR23299:SF50)	*MT1M*	3.11	2.49
Q09666	NEUROBLAST DIFFERENTIATION-ASSOCIATED PROTEIN AHNAK (PTHR23348:SF41)	*AHNAK*	9.48	0.31
Q63HN8	E3 UBIQUITIN-PROTEIN LIGASE RNF213 (PTHR22605:SF16)	*RNF213*	10.35	8.25
O95786	ATP-DEPENDENT RNA HELICASE DDX58-RELATED (PTHR14074:SF16)	*DDX58*	11.21	10.63
P69905	HEMOGLOBIN SUBUNIT ALPHA (PTHR11442:SF48)	*HBA1*	23.19	34.96
Q13113	PDZK1-INTERACTING PROTEIN 1 (PTHR15296:SF0)	*PDZK1IP1*	27.08	38.44
P58546	MYOTROPHIN (PTHR24189:SF52)	*MTPN*	67.35	50.64
	*calcium-binding protein*			
P0DP23	CALMODULIN-1	*CALM1*	0.43	0.43
P09525	ANNEXIN A4 (PTHR10502:SF28)	*ANXA4*	0.86	0.19
P04083	ANNEXIN A1 (PTHR10502:SF17)	*ANXA1*	2.16	3.23
	*scaffold/adaptor protein*			
Q9Y4E1	WASH COMPLEX SUBUNIT 2A-RELATED (PTHR21669:SF1)	*WASHC2C*	0.01	0.01
P43487	RAN-SPECIFIC GTPASE-ACTIVATING PROTEIN (PTHR23138:SF135)	*RANBP1*	0.45	0.03
	*transfer/carrier protein*			
P02753	RETINOL-BINDING PROTEIN 4 (PTHR11873:SF2)	*RBP4*	0.15	0.52
O15127	SECRETORY CARRIER-ASSOCIATED MEMBRANE PROTEIN 2 (PTHR10687:SF7)	*SCAMP2*	0.46	0.19
P80188	NEUTROPHIL GELATINASE-ASSOCIATED LIPOCALIN (PTHR11430:SF13)	*LCN2*	4.76	5.59
	*protease inhibitor*			
P01011	ALPHA-1-ANTICHYMOTRYPSIN (PTHR11461:SF145)	*SERPINA3*	0.39	0.31
P18065	INSULIN-LIKE GROWTH FACTOR-BINDING PROTEIN 2 (PTHR11551:SF5)	*IGFBP2*	0.44	0.01
P17936	INSULIN-LIKE GROWTH FACTOR-BINDING PROTEIN 3 (PTHR11551:SF3)	*IGFBP3*	2.39	2.03
P01024	COMPLEMENT C3 (PTHR11412:SF81)	*C3*	7.03	7.25
P05120	PLASMINOGEN ACTIVATOR INHIBITOR 2 (PTHR11461:SF61)	*SERPINB2*	8.1	7.19
P48594	SERPIN B4 (PTHR11461:SF186)	*SERPINB4*	100	106

**Table 2 cells-11-02465-t002:** Top 25 canonical pathways derived from the IPA analysis.

Ingenuity Canonical Pathways	−log(*p*-Value)	z-Score	Molecules
Antigen Presentation Pathway	11.2		B2M, HLA-B, HLA-C, HLA-DRA, HLA-DRB1, PSMB8, PSMB9, TAP1, TAP2, TAPBP
Phagosome Maturation	10.2		B2M, CTSS, HLA-B, HLA-C, HLA-DRA, HLA-DRB1, PRDX1, PRDX2, PRDX5, PRDX6, TAP1, TSG101, TUBA1A, TUBA4B, TUBG2
NRF2-mediated Oxidative Stress Response	8.8	1.89	ACTA1, ACTC1, ACTG1, GSTA1, GSTA2, HMOX1, MAP2K3, MGST2, NQO1, PRDX1, RALA, RALB, SOD2, SQSTM1, TXN
Protein Ubiquitination Pathway	7.47		B2M, HLA-B, HLA-C, HSPA5, PSMB10, PSMB8, PSMB9, PSMD10, PSME1, PSME2, SASS6, TAP1, TAP2, UBD, UBE2L6, UCHL1
Acute Phase Response Signaling	7.25	2.111	C3, CFB, CP, FN1, HMOX1, MAP2K3, NFKB2, RALA, RALB, RBP4, SERPINA3, SOD2, TTR
Interferon Signaling	7.09	2.646	IFIT3, IFITM1, ISG15, MX1, PSMB8, STAT1, TAP1
EIF2 Signaling	6.12		ACTA1, ACTC1, EIF4E, HSPA5, RALA, RALB, RPLP2, RPS20, RPS28, RPS3, RPS6, RPS9, WARS1
Crosstalk between Dendritic Cells and Natural Killer Cells	5.36		ACTA1, ACTC1, ACTG1, HLA-B, HLA-C, HLA-DRA, HLA-DRB1, NFKB2
Agranulocyte Adhesion and Diapedesis	5.19		ACTA1, ACTC1, ACTG1, CCL20, CXCL1, CXCL10, CXCL2, CXCL8, FN1, ICAM1, MYL6
Glycolysis I	5.18	0.447	BPGM, GAPDH, PGK2, PKM, TPI1
Virus Entry via Endocytic Pathways	4.76		ACTA1, ACTC1, ACTG1, B2M, HLA-B, HLA-C, RALA, RALB
Neuroinflammation Signaling Pathway	4.74	3.051	B2M, CXCL10, CXCL8, HLA-B, HLA-C, HLA-DRA, HLA-DRB1, HMOX1, ICAM1, NFKB2, PLA2G1B, SOD2, STAT1
Type I Diabetes Mellitus Signaling	4.65		CPE, HLA-B, HLA-C, HLA-DRA, HLA-DRB1, MAP2K3, NFKB2, STAT1
Role of IL-17A in Arthritis	4.63		CCL20, CXCL1, CXCL2, CXCL8, MAP2K3, NFKB2
FAT10 Signaling Pathway	4.49		PSME1, PSME2, SQSTM1, UBD
Regulation of eIF4 and p70S6K Signaling	4.36	1	EIF4E, PPP2R1B, RALA, RALB, RPS20, RPS28, RPS3, RPS6, RPS9
Activation of IRF by Cytosolic Pattern Recognition Receptors	4.29	1.633	DDX58, IFIH1, IFIT2, ISG15, NFKB2, STAT1
Sirtuin Signaling Pathway	4.21	0.333	ATG5, ATP5F1B, BPGM, CXCL8, MT-ATP6, NAMPT, NFKB2, NQO1, SOD2, TUBA1A, TUBA4B, UQCRFS1
Communication between Innate and Adaptive Immune Cells	4.16		B2M, CXCL10, CXCL8, HLA-B, HLA-C, HLA-DRA, HLA-DRB1
mTOR Signaling	4.10	0.816	EIF4E, HMOX1, PPP2R1B, RALA, RALB, RPS20, RPS28, RPS3, RPS6, RPS9
Remodeling of Epithelial Adherens Junctions	4.10		ACTA1, ACTC1, ACTG1, MAPRE2, RALA, TUBA1A
Mitochondrial Dysfunction	4.08		ATP5F1B, ATP5PO, COX7C, MT-ATP6, PRDX3, PRDX5, SOD2, UQCRFS1, XDH
Caveolar-mediated Endocytosis Signaling	3.93		ACTA1, ACTC1, ACTG1, B2M, HLA-B, HLA-C
IL-17A Signaling in Gastric Cells	3.90		CCL20, CXCL1, CXCL10, CXCL8
Xenobiotic Metabolism General Signaling Pathway	3.87	−0.707	GSTA1, GSTA2, HMOX1, MAP2K3, MGST2, NQO1, RALA, RALB

**Table 3 cells-11-02465-t003:** Top 25 upstream regulators derived from IPA analysis of differentially expressed proteins obtained in the cytokines vs. control comparison.

Upstream Regulator	Molecule Type	Predicted Activation State	Activation z-Score	*p*-Value of Overlap	Target Molecules in Dataset
IFNG	cytokine	Activated	6.4	1.27 × 10^−27^	ATG5, B2M, BST2, C3, CCL20, CFB, CP, CTSS, CXCL1, CXCL10, CXCL2, CXCL8, DDX58, DIABLO, DUOX2, EEF1A1, ERAP2, FN1, GBP1, GBP2, GBP4, GBP5, HLA-B, HLA-C, HLA-DRA, HLA-DRB1, HMOX1, HSPA5, ICAM1, IDI1, IDO1, IFI30, IFIH1, IFIT2, IFIT3, IFITM1, ISG15, ISG20, LCN2, MIF, MSH2, MX1, NAMPT, NDRG4, NFKB2, NQO1, OAS3, PIGR, PKM, PML, PRDX2, PSMB10, PSMB8, PSMB9, PSME1, PSME2, SCG5, SOD2, SQSTM1, STAT1, TAP1, TAP2, TAPBP, TNFAIP2, TPI1, TYMP, UBD, UBE2L6, WARS1, ZNF638
IRF1	transcription regulator	Activated	3.6	1.44 × 10^−26^	B2M, CCL20, CFB, CTSS, CXCL10, CXCL2, CXCL8, DDX58, GBP2, IDO1, IFIH1, IFIT2, IFIT3, IFITM1, ISG15, MX1, OAS3, PCNA, PIGR, PML, PSMB10, PSMB8, PSMB9, PSME1, PSME2, STAT1, TAP1, TAP2, TAPBP, UBD
STAT1	transcription regulator	Activated	5.4	3.13 × 10^−26^	B2M, BST2, C3, CCL20, CFB, CTSS, CXCL10, CXCL2, CXCL8, DUOX2, GBP1, GBP2, GBP4, GBP5, ICAM1, IDO1, IFI30, IFIH1, IFIT2, IFIT3, IFITM1, ISG15, LCN2, MX1, OAS3, PSMB10, PSMB8, PSMB9, PSME1, PSME2, RNF213, SERPINA3, SERPINB4, STAT1, TAP1, TYMP, UBD, WARS1
TNF	cytokine	Activated	5.9	1.43 × 10^−24^	ACTA1, ANXA1, B2M, BPGM, BST2, C3, CCL20, CFB, CLU, COTL1, CP, CTSS, CXCL1, CXCL10, CXCL2, CXCL8, DDX58, EFHD2, FN1, GBP1, GBP2, GBP4, GSTA1, HID1, HLA-B, HLA-DRA, HMOX1, HSPA5, ICAM1, IDO1, IFIH1, IFIT3, IFITM1, IGFBP2, IGFBP3, ISG15, LCN2, MAP2K3, MGST2, MIF, MSH2, MX1, MYL6, NAMPT, NFKB2, NNMT, NQO1, OAS3, P4HB, PCNA, PIGR, PKM, PML, PPP2R1B, PSMB10, PSMB8, PSMB9, PSME1, PSME2, RPS3, SERPINA3, SERPINB2, SOD2, SQSTM1, STAT1, TAGLN, TAP1, TAPBP, TGM2, TNFAIP2, TXN, TYMP, UBD, VIM, XDH
OSM	cytokine	Activated	4.3	1.33 × 10^−23^	AKR1C1/AKR1C2, ANXA1, B2M, CCL20, CXCL1, CXCL10, CXCL2, CXCL8, FN1, GBP1, GBP2, HLA-B, HLA-C, HMOX1, HSPA5, ICAM1, IGFBP3, ISG20, LCN2, MX1, NAMPT, PCNA, PDCD4, PDZK1IP1, PIGR, PRDX2, PSMB8, PSMB9, REG3A, SERPINA3, SERPINB4, SNRPE, STAT1, TAP1, TAP2, TAPBP, TNFAIP2, TUBA1A, TYMP, UBE2L6, VIM, XDH
IFNA2	cytokine	Activated	4.8	4.02 × 10^−23^	ANXA1, B2M, BST2, CXCL10, DDX58, DIABLO, GBP1, GBP2, GBP4, HLA-B, HLA-C, IDO1, IFIH1, IFIT2, IFIT3, IFITM1, ISG15, ISG20, MTATP6, MT1M, MX1, OAS3, PLA2G1B, PML, PSME1, STAT1, TAP1, TGM2, UBD, UBE2L6
PML	transcription regulator	Activated	2.9	7.25 × 10^−23^	ACTG1, ANXA4, BST2, CPE, CXCL1, HLA-DRA, HMOX1, IDI1, IFIH1, IFIT3, IFITM1, ISG15, ISG20, MX1, NQO1, OAS3, PML, PRDX1, PSMB8, PSMB9, SQSTM1, STAT1, TAP1, TAP2, TXN, VIM
IL1B	cytokine	Activated	5.2	2.81 × 10^−23^	ANXA1, B2M, C3, CCL20, CFB, CP, CTSS, CXCL1, CXCL10, CXCL2, CXCL8, EIF4E, FN1, GBP1, GBP2, GSTA1, GSTA2, HLA-DRA, HMOX1, HSPA5, ICAM1, IDO1, IFIT3, IGFBP3, ISG15, ISG20, LCN2, MIF, MX1, NAMPT, NFKB2, NQO1, P4HB, PCNA, PIGR, PSMB10, PSMB8, PSMB9, PSME2, RAN, SERPINA3, SERPINB2, SOD2, STAT1, TAP2, TGM2, TNFAIP2, TYMP, UBD, UBE2L6, VIM, XDH
IFNL1	cytokine	Activated	4.4	1.84 × 10^−21^	BST2, CXCL10, CXCL8, DDX58, GBP1, GBP5, HLA-B, HLA-C, IFIH1, IFIT2, IFIT3, IFITM1, ISG15, ISG20, MX1, OAS3, PML, PSMB9, STAT1, UBE2L6
Interferon alpha	group	Activated	5.4	6.66 × 10^−21^	APOL2, B2M, BST2, C3, CXCL1, CXCL10, CXCL2, CXCL8, DDX58, GBP1, GBP2, GBP5, HLA-B, HLA-C, ICAM1, IDO1, IFIH1, IFIT2, IFIT3, IFITM1, ISG15, ISG20, LAP3, MX1, OAS3, PML, PSMB8, PSMB9, RNF213, STAT1, TAP1, TAP2, TAPBP, TYMP, UBE2L6, WARS1
CD40LG	cytokine	Activated	3.2	8.58 × 10^−21^	B2M, CCL20, CLU, CXCL1, CXCL10, CXCL2, CXCL8, ICAM1, IDO1, IFIT2, IFIT3, IFITM1, ISG15, LMAN2, MAP2K3, MIF, MSH2, MX1, NAMPT, NFKB2, PML, PSMB10, PSMB8, PSMB9, PSME1, PSME2, SOD2, STAT1, TAP1, TAP2, TGM2, TNFAIP2, TYMP, UBD
RELA	transcription regulator	Activated	2.4	1.77 × 10^−20^	B2M, C3, CCL20, CFB, CXCL1, CXCL10, CXCL2, CXCL8, ERAP2, FN1, GBP1, GSTA1, GSTA2, HLA-B, HMOX1, ICAM1, IGFBP2, ISG15, MIF, NAMPT, NFKB2, PKM, PRDX6, PSMB10, PSMB9, REG3A, SOD2, TAP1, TAP2, TAPBP, TGM2, TNFAIP2, TPMT, UBD, VIM
IRF7	transcription regulator	Activated	4.7	5.05 × 10^−20^	CXCL10, DDX58, GBP1, GBP4, GBP5, IDO1, IFIH1, IFIT2, IFIT3, IFITM1, ISG15, ISG20, MX1, NAMPT, OAS3, PSMB10, PSMB8, PSMB9, PSME1, PSME2, STAT1, TAP1, TAP2, UBE2L6
APP	other	Activated	3.6	1.64 × 10^−19^	ACTA1, ACTG1, ATP1A3, ATP5F1B, C3, CCL20, CLU, CP, CXCL1, CXCL10, CXCL2, CXCL8, DDX58, FN1, GAPDH, GBP2, GBP4, HMOX1, HSPA5, ICAM1, IDO1, IFIH1, IFIT2, IGFBP2, ISG20, MT-ATP6, NAMPT, PDCD4, PGK2, PKM, PRDX2, PRDX5, PRDX6, PSME1, RAN, RNF213, RPS6, SCG5, SOD2, SPTAN1, SQSTM1, TAGLN, TNFAIP2, TPI1, TTR, TUBA1A, TXN, UCHL1
Ifnar	group	Activated	4.3	3.37 × 10^−19^	B2M, C3, CXCL10, DDX58, GBP2, IDO1, IFIH1, IFIT2, IFIT3, ISG15, ISG20, PSMB8, PSMB9, RNF213, STAT1, TAP1, TAP2, TAPBP, UBE2L6
MYC	transcription regulator		−0.8	5.15 × 10^−18^	ACAT1, ACTA1, ANXA4, ARF3, BOP1, CLU, CXCL10, CXCL8, DDX21, DEK, EIF4E, FN1, GAPDH, GBP2, GBP4, H2AZ1, HDAC2, HLA-B, HMOX1, ICAM1, IFIH1, IFIT2, IFIT3, ISG20, LIMS1, MIF, MRPL12, MSH2, MX1, MYO1B, NQO1, PCNA, PDCD4, PKM, PML, PRDX2, PRDX3, PSMB8, RANBP1, RPLP2, RPS20, RPS28, RPS3, RPS6, RPS9, SOD2, SQOR, STAT1, TMEM97, TPI1, TXN, VIM
TP53	transcription regulator		0.9	8.64 × 10^−18^	ACAT1, ANXA1, ANXA4, BOP1, CCS, CLU, CP, CXCL1, CXCL10, CXCL8, DEK, EEF1A1, FN1, GAPDH, GBP1, GPRC5C, H2AZ1, HDAC2, HLA-B, HMOX1, HSPA5, ICAM1, IFI30, IGFBP2, IGFBP3, ISG15, LMAN2, MAP2K3, MAPRE2, MDH2, MGST2, MRPL12, MRPL46, MSH2, MX1, NAMPT, NDRG4, NFKB2, P4HB, PCNA, PFN1, PKM, PML, PPM1B, PRDX2, PRDX3, PRDX6, PSMB9, RAN, RPS20, RPS3, SEC62, SERPINA3, SERPINB2, SOD2, SQOR, STAT1, TAP1, TAP2, TGM2, TMEM97, TNFAIP2, TSG101, UBD, UQCRFS1, VIM
STAT3	transcription regulator		1.0	1.29 × 10^−17^	BST2, CCL20, CFB, CXCL10, CXCL2, CXCL8, FN1, GBP2, GBP5, HMOX1, ICAM1, IFI30, IFIH1, IFIT2, IFIT3, IFITM1, ISG15, ISG20, MT-ATP6, MX1, NAMPT, NFKB2, OAS3, PML, PSMB8, PSMB9, REG1A, REG3A, SERPINA3, SERPINB4, SOD2, STAT1, TAGLN, TAP1, UBD, VIM, WARS1
IFNAR2	transmembrane receptor	Activated	3.0	3.31 × 10^−17^	CXCL10, DDX58, GBP4, IDO1, IFIH1, IFITM1, ISG15, PSMB10, PSMB8, PSMB9, PSME2, TGM2, UBD, UBE2L6
HRAS	enzyme		1.4	1.89 × 10^−16^	ACTA1, ACTG1, ATP1A3, ATP5F1B, B2M, CFB, CHGA, CLU, CTSS, CXCL1, CXCL10, CXCL8, FN1, HLA-B, HMOX1, ICAM1, IFI30, MRPL12, NQO1, P4HB, PCNA, PDCD4, PFN1, PML, PRDX2, PRDX6, PSMB9, RPS3, RPS6, SERPINA3, SERPINB2, TAGLN, TAP1, TAP2, TGM2, TYMP, VIM
MAPK1	kinase	Inhibited	−4.8	2.18 × 10^−16^	BST2, CFB, CXCL8, DDX58, FN1, GBP1, GBP2, GBP5, HLA-B, HLA-C, HMOX1, IFI30, IFIH1, IFIT2, IFIT3, IFITM1, ISG15, ISG20, LAP3, OAS3, PML, PSMB8, PSMB9, PSME2, STAT1, TAP1, TRIM25, UBE2L6, VIM
IL6	cytokine	Activated	2.5	2.50 × 10^−16^	AHNAK, ANXA1, BST2, C3, CCL20, CLU, CP, CXCL1, CXCL10, CXCL2, CXCL8, FN1, GBP2, GCG, GSTA1, HMOX1, HSPA5, ICAM1, IDO1, IFIT2, IGFBP3, IK, LCN2, NAMPT, PCNA, PSMB10, PSMB8, PSMB9, PSME1, PSME2, REG1A, SERPINA3, SOD2, SST, STAT1, TAP1, TGM2, TTR, VIM, WARS1
IRF3	transcription regulator	Activated	3.7	4.26 × 10^−16^	AHNAK, ANXA4, B2M, CXCL1, CXCL10, CXCL8, DDX58, FN1, GBP1, GBP5, IFIH1, IFIT2, IFIT3, ISG15, ISG20, LCN2, OAS3, PML, STAT1, TAP1, UBE2L6, VIM
PRL	cytokine	Activated	2.3	6.90 × 10^−16^	ARF3, B2M, BST2, CCL20, CLU, CTSS, CXCL10, DDX58, FN1, IFIH1, IFIT3, IFITM1, IGFBP3, ISG15, OAS3, P4HB, PCNA, PSME1, PSME2, RALB, SERPINA3, STAT1, TRIM25, VIM, XDH
IRF2	transcription regulator		0.9	2.55 × 10^−15^	B2M, CFB, CTSS, CXCL10, GBP1, HLA-B, ISG15, LCN2, NAMPT, PSMB10, PSMB8, PSMB9, PSME1, PSME2, TAP1, TAP2, TAPBP, UBE2L6

## Data Availability

All relevant data have been provided in the main text and the Appendix A.

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
