# Peer review of "The Protective Action of Metformin against Pro-Inflammatory Cytokine-Induced Human Islet Cell Damage and the Mechanisms Involved"

_cells, 2022, doi:10.3390/cells11152465_

Round 1

Reviewer 1 Report

The present study aims to analyze the effect of metformin,  one of the mostly prescribed anti-diabetic drug, in  the protection of human beta cells from inflammatory cytokines. To this aim, the authors first analysed the effect of metformin in glucose-stimulated insulin secretion and casp3/7 activation. Next they performed a MS-based proteomic analysis in human islets treated with cytokines, cytokines and metformin or left untreated, as control.

The present study suggests that metformin, besides in T2D, might be considered for beta cell protection in other types of diabetes, possibly including early T1D.

The paper is interesting and novel, the story is well-written and the figures are nice. Unfortunately the proteomic analysis has been performed in single samples, without biological replicates. This didnt'allow the author to perform a proper statistical analysis to identify differentially regulated proteins. Surprinsigly GO-enrichment analysis didn't reveal significant differences in the modulation of almost all the identified GO terms (Fig. 4A). I would suggest the authors to perform heatmaps with enrichment score, instead of pValues, since this could help to detect any differences between the experimental conditions, if any. A number of terms are poorly related to diabetes and islets (eg. SARS Cov pathway, influenza or bacterial infections). Can the authors comment on this? Additionally,  interleukin signaling pathways as well as inflammatory pathways (fig. 5a) were found to be equally enriched in the two experimental conditions. However, the author proposed metformin as a protective agent in inflammatory response of islets. Can the author comment on this point? Additionally, in Fig. 5A, it is not clear how the most significantly affected canonical pathways were selected. 

Finally, besides these minor points, the paper is novel and interesting and I endorse it for publications.

Reviewer 2 Report

The protective action of metformin against pro-inflammatory cytokine-induced human islet cell damage and the mechanisms involved

General comments

The study assesses whether metformin can relieve human β-cells stress induced by pro-inflammatory cytokines in type-1 diabetes and investigated the mechanisms involved using shotgun proteomics.

The study reports that metformin prevented part of the deleterious actions of pro-inflammatory cytokines on human β cells which was accompanied by islet proteome modifications.

The paper is well written. The authors can address the specific concerns below to improve the paper further.

Specific comments

Abstract & Introduction

The abstract concisely summarizes the study.

The introduction provides satisfactory background on diabetes, how it is treated, the known mechanisms and presents the gap that the study attempts to address.

Methods

What was the rationale for selecting multi-organ donors aged 71±9 years with BMI of 26±3 kg/m2?

Provide the ethics committee number.

Why was one-dimension electrophoresis (1DE) used while it’s known that 2DE is well known for efficiently separating proteins, their variants and modifications (up to 15000 proteins)?

Results, Discussion and Conclusion

- The figures are well presented with descriptive legends that make it easy to follow.

-Tables 1,2 and 3 can be moved to the supplementary section.

-Back up the statements by including the precise statistical significance value that was observed.

-The authors should refrain from including result figures in the discussion section e.g. Figure1, Table S2, etc.

-The authors should mention the limitations of the current study (especially the technical challenges). After mentioning those limitations, the authors should proceed to assure the reader that how results of the study are still credible with the stated limitations.

-The conclusion is appropriate as per the findings of the study.

References

-The references are appropriate.
